# Reduction in Phosphoribulokinase Amount and Re-Routing Metabolism in *Chlamydomonas reinhardtii* CP12 Mutants

**DOI:** 10.3390/ijms23052710

**Published:** 2022-02-28

**Authors:** Cassy Gérard, Régine Lebrun, Erwan Lemesle, Luisana Avilan, Kwang Suk Chang, EonSeon Jin, Frédéric Carrière, Brigitte Gontero, Hélène Launay

**Affiliations:** 1Aix Marseille Univ, CNRS, BIP, UMR 7281, IMM, FR3479, 31 Chemin J. Aiguier, CEDEX 9, 13402 Marseille, France; cgerard@imm.cnrs.fr (C.G.); erwan.lemesle@etu.univ-amu.fr (E.L.); carriere@imm.cnrs.fr (F.C.); 2Plate-forme Protéomique, Marseille Protéomique (MaP), IMM FR 3479, 31 Chemin Joseph Aiguier, 13009 Marseille, France; rlebrun@imm.cnrs.fr; 3Centre for Enzyme Innovation, School of Biological Sciences, Institute of Biological and Biomedical Sciences, University of Portsmouth, Portsmouth PO1 2DY, UK; luisana.avilan@port.ac.uk; 4Department of Life Science, Research Institute for Natural Sciences, Hanyang University, Seoul 04763, Korea; kschang@hanyang.ac.kr (K.S.C.); esjin@hanyang.ac.kr (E.J.)

**Keywords:** Calvin–Benson–Bassham, CP12, enzymatic activity, intrinsically disordered protein, quantitative proteomics, photosynthesis, glyoxylate pathway, carbon allocation, nitrogen uptake

## Abstract

The chloroplast protein CP12 is involved in the dark/light regulation of the Calvin–Benson–Bassham cycle, in particular, in the dark inhibition of two enzymes: glyceraldehyde−3-phosphate dehydrogenase (GAPDH) and phosphoribulokinase (PRK), but other functions related to stress have been proposed. We knocked out the unique *CP12* gene to prevent its expression in *Chlamydomonas reinhardtii* (ΔCP12). The growth rates of both wild-type and ΔCP12 cells were nearly identical, as was the GAPDH protein abundance and activity in both cell lines. On the contrary, the abundance of PRK and its specific activity were significantly reduced in ΔCP12, as revealed by relative quantitative proteomics. Isolated PRK lost irreversibly its activity over-time in vitro, which was prevented in the presence of recombinant CP12 in a redox-independent manner. We have identified amino acid residues in the CP12 protein that are required for this new function preserving PRK activity. Numerous proteins involved in redox homeostasis and stress responses were more abundant and the expressions of various metabolic pathways were also increased or decreased in the absence of CP12. These results highlight CP12 as a moonlighting protein with additional functions beyond its well-known regulatory role in carbon metabolism.

## 1. Introduction

In all photosynthetic eukaryotes, the Calvin–Benson–Bassham (CBB) cycle involved in CO_2_ assimilation is dependent upon reducing power and ATP production from the photochemical phase of photosynthesis. In the absence of light, the CBB cycle is inhibited; a fine regulation of CBB enzymes is achieved by post-translational redox modifications of regulatory cysteine residues [1,2,3], metabolite concentrations and pH transition in the chloroplast stroma [4]. Protein–protein interactions, involving the CP12 protein and two enzymes belonging to this cycle, namely the phosphoribulokinase (PRK) and the glyceraldehyde−3-phosphate dehydrogenase (GAPDH), also regulate the CBB cycle [5,6,7,8].

CP12 is a small chloroplast protein of 8.5 kDa, encoded by the nuclear genome, that is widespread in photosynthetic organisms, such as higher plants, diatoms, and red and green algae, as well as cyanobacteria [9]. This protein has a pair of cysteine residues at the C-terminus and, in Viridiplantae, a second pair at the N-terminal extremity. In the dark, CP12 is oxidized and its cysteine residues form disulfide bridges. In the light, CP12 is reduced, and its disulfide bridges are disrupted. The reduction in these disulfide bridges triggers conformational changes and leads to a fully disordered CP12, while, under its oxidized state, CP12 oscillates between a disordered and a partly ordered state [10,11]. This redox-dependent conditional disorder, also described for other proteins such as Cox17 and Hsp33 [12], allows the regulation of the CBB cycle and, in particular, of PRK and GAPDH activities. Oxidized CP12 sequentially binds to GAPDH and then PRK [7]. The two enzymes are inhibited within this ternary complex and, thus, the CBB cycle becomes inactive. Upon CP12 reduction, the two enzymes are released from the supramolecular complex and become fully active, thus, allowing the CBB cycle to function.

CP12, similar to many disordered proteins, appears as a moonlighting protein with other functions besides its important regulatory role in the CBB cycle [13]. In cyanobacteria, CP12 was shown to bind NADH and NADPH [14]. In *Chlamydomonas reinhardtii*, CP12 acts as a specific chaperone, being able to protect GAPDH against heat-induced inactivation and aggregation [15] and can bind Cu^2+^ and Ni^2+^ [16,17], while, in the higher plant *Arabidopsis thaliana*, it can bind Ca^2+^ [18]. In *C. reinhardtii*, CP12 can interact with a large number of partners, such as malate dehydrogenase, elongation factor 1α2 and 38 kDa ribosome-associated protein but to a lesser extent than with PRK, GAPDH and another enzyme from the CBB cycle, the aldolase [19]. The function of CP12 in vivo is, thus, broader than initially envisaged. However, the use of approaches to decrease the amount or to delete the expression of CP12 to elucidate its global function has been initiated only on a few photosynthetic organisms. To the best of our knowledge, today, only CP12-transgenic antisense in the tobacco, *Nicotiana tabacum*, in *A. thaliana* plants, in the tropical legume *Stylosanthes guianensis* and a cyanobacterial CP12 insertional mutant (knockout mutant of *Synechococcus elongatus* PCC7942) have been studied [14,20,21,22,23].

However, no data are available for green algae; therefore, the aim of the work presented here was to investigate further the role of CP12 in vivo by suppressing its gene expression in *C. reinhardtii*. To achieve this, knock-out mutants of the *CP12* gene coding the unique isoform of CP12 in the cell-wall deficient *C. reinhardtii* line CC4349 were constructed using CRISPR-Cas9 ribonucleoprotein (RNP)-mediated method. We first evaluated the impact of CP12 deletion on the growth rates and PRK and GAPDH activities in cell extracts. We completed these targeted investigations with a global untargeted quantitative proteomic analysis to evaluate the effect of CP12 deletion on the whole *C. reinhardtii* proteome. In parallel, the effect of CP12 on the protection of PRK activity against irreversible damage was monitored in vitro; the amino acid residues in CP12 involved in this protection were identified using CP12 protein mutants.

## 2. Results

### 2.1. The Absence of CP12 Protein Has No Effect on C. reinhardtii Growth

The genomic sequence of knock-out mutants (ΔCP12-Cr) revealed that the gRNA binding site of *CP12* was mutated. This mutation resulted in an early stop codon induced by co-transformations of CRISPR-Cas9-RNP and the aminoglycoside phosphotransferase from *Streptomyces hygroscopicus* (*aph7*) gene expression cassette at the target site (Appendix A).

Once the targeted mutation of the *CP12* gene was confirmed, we tested the growth of wild-type (WT) CC4349 *C. reinhardtii* (named WT-Cr hereafter) and ΔCP12-Cr strains. Under continuous light or under a dark/light cycle, the growth curves of WT-Cr and ΔCP12-Cr were similar (Figure 1), indicating that the absence of the CP12 protein did not impair the growth of *C. reinhardtii*.

### 2.2. Absence of CP12 Abolishes GAPDH Regulation

In order to determine the effect of the absence of CP12 protein on the GAPDH activity, we measured, continuously, the NADPH-dependent activity of GAPDH in cell extracts from WT-Cr and ∆CP12-Cr strains after their reduction with dithiothreitol (DTT). The steady-state activity of GAPDH was 212 ± 73 µM·min^−1^ per mg of protein in the cell extract of WT-Cr and 332 ± 47 µM·min^−1^ per mg of protein in the cell extract of ΔCP12-Cr (Table 1). The NADPH-dependent steady-state activity of GAPDH was slightly higher (1.5-fold) in the ΔCP12-Cr than in WT-Cr.

When the activity of GAPDH enzyme from the untreated WT-Cr cell extract was continuously monitored, the progress curve displayed a lag phase of about 1 min and a linear phase that corresponds to an activity of 168 ± 47 µM·min^−1^·mg^−1^ of proteins (Figure 2a and Table 1). The activity of the enzyme of the untreated samples was slightly lower (1.3-fold) than the one obtained with a pre-treatment of the cell extracts with 20 mM DTT (Table 1).

In contrast, in the ∆CP12-Cr cell extracts, the NADPH-dependent activity of GAPDH displayed a linear curve regardless of DTT (Table 1 and Figure 2a).

### 2.3. PRK Activity Is Lower in ΔCP12-Cr Compared to WT-Cr

We also measured, continuously, the PRK activity in cell extracts that were pre-treated with 20 mM DTT; the activity was 1143 ± 328 µM·min^−1^ per mg of protein in the cell extract of WT-Cr and 430 ± 94 µM·min^−1^ per mg of protein in the cell extract of ΔCP12-Cr (Table 1). The steady-state PRK activity in ∆CP12-Cr cell extracts was 3-fold lower than in WT-Cr cell extracts (Student *t*-test, *p* = 0.0004).

As observed for GAPDH, the activity of PRK in the untreated WT-Cr sample displayed a lag (Figure 2b). The average activity PRK was 940 ± 309 µM·min^−1^ per mg of protein in the cell extract (Table 1). This activity was slightly lower than the activity measured on reduced WT-Cr cell extract (1.2-fold).

In the untreated ∆CP12-Cr cell extracts, PRK activity progress curves also displayed a lag slightly higher (about 3 min) than the lag observed in the WT-Cr strain (about 2 min) (Figure 2b).

### 2.4. Lower PRK Abundance in ΔCP12-Cr Compared to WT-Cr

To get a more complete view of the differentially expressed proteins, in addition to PRK and GAPDH, we used untargeted quantitative proteomics with three biological replicates of ΔCP12-Cr and WT-Cr cell extracts, and two technical replicates. A total of 4170 proteins could be identified, from which 2524 proteins were kept for the quantitative analysis (Materials and Methods). Most of these proteins were in similar amount in ΔCP12-Cr and WT-Cr (Figure 3 and Appendix A), as evidenced by the quantification of their peptides. However, applying a two-sample *t*-test with a permutation-based false discovery rate (FDR) set at 0.01, the difference in abundance of 130 proteins was observed as statistically significant; 95 proteins were more abundant and 35 proteins less abundant in ∆CP12-Cr than in WT-Cr strains.

As a first analysis, we looked for the relative amount of GAPDH (Cre01.g010900) and PRK (Cre12.g554800) in the ΔCP12-Cr and WT-Cr strains. No significant difference was observed for the GAPDH amount, whereas a significant difference was observed for PRK amount, with a 6.5-fold decrease of PRK amount in ΔCP12-Cr compared to the WT-Cr strain (Figure 3a). Other enzymes of the CBB cycle, the fructose−1,6-bisphosphatase 2 (FBP−2, Cre12.g510650) and the RuBisCO large subunit methyltransferase (Cre09.g388500) were both slightly reduced (1.7-fold and 1.8-fold, respectively) in the ΔCP12-Cr compared to the WT-Cr strain. In contrast, the EPYC1 (essential pyrenoid component 1) protein (Cre10.g436550), was more abundant (4.9-fold) in the ΔCP12-Cr compared to the WT-Cr strain.

The abundances that were the most decreased correspond to proteins involved in the photochemical phase of photosynthesis. For example, the chlorophyll a/b binding protein of photosystem II (Cre17.g720250) decreased 375-fold. Proteins involved in electron transfer from photosynthesis to CBB cycle were also reduced, such as the ferredoxin NADP^+^ reductase (Cre11.g476750, 1.65-fold), the chloroplast ferredoxin (Cre14.g626700, 0.5-fold) and the thioredoxin F1 (Cre05.g243050, 2.26-fold).

### 2.5. Differential Levels of Proteins Associated with Various Pathways in ΔCP12-Cr vs. WT-Cr

The deletion of CP12 induced changes in the abundance of proteins beyond those involved in the photosynthesis pathways. Relative quantitative proteomic analysis revealed that, in ΔCP12-Cr cells, other metabolic pathways were affected, in particular, carbohydrate-related pathways (Figure 3b). Key proteins involved in gluconeogenesis were more abundant, such as transaldolase (Cre01.g032650, 6-fold) and the fructose−1,6-bisphosphatase 1, FBP−1 (Cre07.g338451, 3.5-fold), whereas pyruvate kinase 1 and 5 (Cre12.g533550 and Cre02.g147900), related to glycolysis, were less abundant in ΔCP12-Cr cells (1.6 and 2.2-fold). The starch synthesis also seemed to be affected, with a decrease in soluble starch synthase I (Cre04.g215150) and in the starch branching enzyme (Cre10.g444700) by, respectively, 2.1- and 1.7-fold. On the contrary, sucrose synthesis seems to be favored since the amount of UDP-glucose pyrophosphorylase (Cre04.g229700) was higher (1.4-fold) in ΔCP12-Cr cells.

Proteins related to the glyoxylate and to the Krebs cycle were significantly more abundant in ΔCP12-Cr cells compared to WT-Cr cells (Figure 3c and Appendix A). The proteins that showed the highest increase fold in the ΔCP12-Cr line compared to WT-Cr line were the isocitrate lyase 1 (Cre06.g282800) and the malate synthase (Cre03.g144807) (270- and 73-fold respectively); together, they belong to the glyoxylate pathway with two other proteins, which are also more abundant [25,26]. The glyoxylate cycle is localized in the glyoxysome (a specialized peroxisome) of *C. reinhardtii*; the peroxisomal biogenesis factor 11 (Cre06.g263300, 2.5-fold) was also more abundant in ΔCP12-Cr cells. Isocitrate lyase and malate synthase belong to the same group of co-expressed genes with an intracellular acetate transporter (Cre17.g700750, 15-fold), which was also significantly more abundant in ΔCP12-Cr cells compared to the WT-Cr cells. Seven enzymes of the Krebs cycle were also more abundant in the ΔCP12-Cr line (increase from 1.4- to 3-fold). With a two-sample *t*-test with permutation-based FDR set at 0.05, two ammonium transporters (Cre03.g159254 and Cre13.g569850) were les abundant (5-fold and 2.9-fold respectively) in ΔCP12-Cr vs WT-Cr cell lines.

There are also proteins involved in a wide range of general cellular processes, such as amino acid metabolism, transcription, and translation, which were affected (Figure 3d and Appendix A). Changes related to stress response occurred in ΔCP12-Cr compared to WT-Cr with an increase of six chaperones and thirteen proteins linked to redox homeostasis and responses to the accumulation of reactive oxygen species (ROS). Several proteins involved in lipid metabolism were differently present. Lastly, the abundance of six proteins (mostly flagellar associated proteins and radial spoke proteins) involved in *C. reinhardtii* motility was lower in ΔCP12-Cr.

### 2.6. CP12 Protein Protects PRK from Irreversible Inactivation In Vitro

Since the absence of CP12 in the ΔCP12-Cr cells was associated with a reduction in PRK amount and activity, we tested in vitro the effect of CP12 on PRK activity. We monitored the activity of PRK upon oxidation, both in the absence and in the presence of CP12. Under aerobic conditions, PRK activity decreased over time faster in the absence of CP12 than in the presence of CP12 (Figure 4a). The inhibition of PRK is triggered by its oxidation through the formation of disulfide bridge between C16 and C55, with a redox mid-potential at pH 7.9 of −333 mV [27]; this oxidation is dependent on the redox potential of the solution. Since CP12 is a redox mediator with two redox mid-potentials (−343 and −356 mV at pH 7.9) lower than that of PRK, we hypothesized that the presence of CP12 in the solution could result in a slower oxidation of the solution and, thus, in a slower inactivation of PRK.

The oxidative inhibition of PRK is reversible upon addition of DTT [28]. However, in the absence of CP12, reduction in the samples by DTT failed to restore the initial activity of PRK (Figure 4b). This suggests that irreversible damage occurred on PRK beyond the formation of the inhibitory and reversible C16-C55 disulfide bridge, and that PRK irreversible inhibition is not linked to its redox state. On the contrary, in the presence of CP12, the addition of DTT restored the initial PRK activity, indicating that CP12 prevented this irreversible inactivation.

### 2.7. Identification of CP12 Residues Involved in PRK Protection

It is known that CP12 interacts with PRK within the ternary GAPDH-CP12-PRK complex [7]; this interaction involves CP12 residues A34 to H47 in higher plants and cyanobacteria based on structural data [29,30,31] (Figure 5b). Using site-directed mutagenesis, we investigated the contribution of the corresponding region and other residues in *C. reinhardtii* CP12 to the protection of PRK against irreversible inactivation. Point mutations on CP12 (W35A, D36K, E39A, E39K, E40K, H47L) and double mutations (E39A/E40A; E39K/E40K) in the consensus core sequence rendered CP12 mutants unable to protect PRK activity; E40A and H47N mutants only partially protected PRK activity, while all other mutations, including mutation of the N-terminal cysteine residue 23, did not change the protective effect of CP12 (Figure 5a).

From the above-mentioned crystallographic structures, these residues are located at the CP12-PRK interface in the ternary complex (Figure 5b). Some conservative mutations such as W35F did not prevent protection, consistent with the large aromatic nature of both W and F side chains. Mutation of the negatively charged D36, E39 and E40 residues into neutral A residues or positively charged K residues resulted in irreversible PRK damage (Figure 5a).

## 3. Discussion

The scope of functions of the CP12 proteins is constantly increasing, supporting their definition as moonlighting proteins. The dark down-regulation of GAPDH and PRK activities by CP12 was confirmed by CP12 disruption mutants in *S. elongatus* PCC7942 grown under dark/light conditions, and by the presence of the supramolecular GAPDH-CP12-PRK complex in dark-treated *S. elongatus* PCC7942 cells but not in light-exposed cells [14]. However, the affinity of CP12 for GAPDH and PRK differs between species, and the GAPDH-CP12-PRK complex stability in dark-treated leaves is weak in *N. tabacum* and *A. thaliana* [33]. In *N. tabacum* and in the tropical legume *S. guianensis,* decreasing the amount of CP12 was associated with a moderate reduction in maximum activity for PRK and GAPDH compared to WT leaves [20,22]. Reversely, a higher level of CP12 in *S. guianensis* was associated with higher maximum activities of GAPDH and PRK [23]. In *A. thaliana*, decreasing the CP12 level only reduced the maximum activity of PRK, while the level of GAPDH activity remained stable [22]. These results puzzle the mechanism by which CP12 regulates GAPDH and PRK.

These and other in vivo studies have also suggested other functions for CP12. A role in the protection against ROS damage caused by high light has been proposed in *S. elongatus* PCC7942 [34]; roles in carbon partitioning, malate valve capacity and redox homeostasis and carbon/nitrogen homeostasis have been proposed in *N. tabacum* [20]; a role in the photosynthesis efficiency has been proposed in *N. tabacum*, *A. thaliana* and *S. guianensis* [21,22,23]; this list is not exhaustive. The numerous in vitro studies also indicated that CP12 could be considered as a moonlighting protein (Introduction) [13].

In higher plants, CP12 proteins are encoded by small multigene families; it is a challenge to decipher the importance of each individual *CP12* gene in vivo [22]. As cyanobacteria, the green alga *C. reinhardtii* possesses a unique isoform of CP12 that eases the investigation of CP12 function in vivo. The growth of *S. elongatus* PCC7942 CP12 deletion mutants (ΔCP12-Sc) under continuous low light (40 µmol photon·m^2^·s^−1^) is not affected by CP12 deletion, but under continuous high light (100 µmol photon·m^2^·s^−1^) or under dark/light cycles, CP12-deleted mutants showed altered growth [14,34]. The cyanobacterial CP12 does not possess the N-terminal disulfide bridge of the canonical CP12; therefore, *C. reinhardtii* having a unique form of CP12 with N-terminal and C-terminal bridges is a better model to study the multiple functions of CP12 [9]. Contrary to what was observed in cyanobacteria, deletion of CP12 in *C. reinhardtii* did not impair growth under continuous high light (120 µmol photons·m^2^·s^−1^). Our quantitative proteomic analysis of ΔCP12-Cr cells indicated, however, that the amount of a large number of proteins involved in ROS protection was higher compared to WT-Cr cells; these proteins are localized in other organelles than the chloroplast. This proteomics analysis is in agreement with the results from Tamoi et al. [34], suggesting that, under 100 µmol photons·m^2^·s^−1^, the mutant cell lines were more susceptible to oxidative damage.

Under phototrophic conditions, in ΔCP12-Cr, we observed a higher amount of enzymes of the glyoxylate pathway that is involved in acetate assimilation and that occurs in the glyoxysome (Figure 6) [26]. Pre-cultures were performed under mixotrophic conditions, where the presence of this pathway is relevant. We had, therefore, quantified acetate in the pre-culture medium after five days by nuclear magnetic resonance (Appendix A); only traces of acetate remained present in the phototrophic culture. We would, thus, anticipate that the glyoxylate pathway should be absent in *C. reinhardtii* WT cells when shifting their culture from mixotrophic to phototrophic conditions. From the presence of the glyoxylate pathway enzymes in ΔCP12-Cr cells, we, therefore, hypothesized that these cells are unable to turn off this cycle in the absence of CP12. Metabolites of this pathway are also involved in the Krebs cycle that occurs in the mitochondrion; enzymes of this pathway are also present in higher amount in ΔCP12-Cr compared to WT-Cr cells. The Krebs cycle metabolites such as malate and fumarate were shown to be significantly higher in *N. tabacum* antisense CP12-disrupted leaves (ΔCP12-Nt), whereas isocitrate and 2-oxoglutarate were significantly lower [20]. These metabolites (except 2-oxoglutarate) are shared by both the Krebs and glyoxylate pathways, and could feed both cycles [35].

Interestingly, in ΔCP12-Nt, the activity of the NADP-malate dehydrogenase (NADP-MDH) was lower than in wild-type plants; however, the protein amount was unchanged [20]. This enzyme is involved in the regulation of the NAD(P)H/NAD(P) ratio; NAD(P)H is known to be an important regulatory metabolite. In particular, the CBB cycle requires a finely tuned energetic balance between NADPH and ATP. CP12 is a conductor of this cycle by regulating the unique reduction step of the CBB, i.e., the NADPH-dependent GAPDH activity, together with one of the two ATP-dependent phosphorylation reactions [1]. In this work, we showed that GAPDH activity displayed a lag in untreated WT-Cr cell extracts, consistent with the presence of an inactive GAPDH-CP12-PRK complex that dissociates into active forms in the reaction mixture, as previously described [7]. In the reduced WT-Cr cell extracts, this lag disappeared, consistent with the dissociation of the complex prior to assay. In the absence of CP12 (ΔCP12-Cr cell extracts), this lag was absent; this is in good agreement with the absence of the ternary complex and the absence of autonomous GAPDH regulation. In the green algae, the unique homotetrameric isoform of GAPDH (A4) in contrast to the heterotetrameric isoform of higher plants (A2B2) does not possess the regulatory cysteine residues; its regulation is strictly dependent upon the formation of the GAPDH-CP12-PRK complex [36].

Similarly, PRK activity displayed a lag in untreated WT-Cr cell extracts. The regulation of PRK is more complex, PRK possesses regulatory cysteine residues C16 and C55 that are targets of thioredoxin F1 [37]; PRK is, thus, redox autonomously regulated. It is also inhibited in the ternary complex. The association of CP12 with PRK was shown to ease its reducing activation; we have confirmed that, in the presence of CP12, PRK activation within the supramolecular complex is faster than in the absence of CP12 [38]. Indeed, in our activity assays, in the absence of CP12, this lag phase lasted longer than in the WT-Cr cell extracts. The regulation of GAPDH and PRK in *C. reinhardtii* WT-Cr and ΔCP12-Cr cell extracts is, thus, consistent with the literature.

The maximum activity of GAPDH was identical in both WT-Cr and ΔCP12-Cr cells, as it was observed in *A. thaliana* CP12-disrupted mutant; this suggests that, even in the absence of its specific chaperone, GAPDH was stable [15]. On the contrary, we observed that both the PRK maximum activity and protein amount were decreased. The growth of ΔCP12-Cr cells was not affected, however, which is consistent with the fact that reduction in photosynthesis in *N. tabacum* PRK antisense plants is only evident in plants with ≤ 20% of wild-type PRK levels [39,40]. Lopez-Calcagno et al. have shown that the low PRK amount was not correlated to a decrease of mRNA; this indicated that the regulation is at the translational or post-translational level. Here, we showed in vitro that PRK activity is irreversibly damaged in the absence of CP12; therefore, CP12 can protect PRK from inactivation. Moreover, our quantitative proteomic results suggest that PRK is also protected by CP12 from degradation under continuous light in WT-Cr, whereas absence of CP12 results in much lower (6-fold) levels of PRK. The mechanism by which PRK is degraded has not been explored yet, but one can speculate that the interaction between PRK and CP12, which exists in the cell, prevents PRK from proteolytic degradation. The absolute concentration of CP12 remains to be determined, but is sufficient for the formation of the ternary complex that we observed in our activity assays. Under continuous light, the reducing power of photosynthesis results in reducing condition for both PRK and CP12. In vitro, we confirmed that the protection of PRK is not dependent on the redox state of CP12 since it was also observed with the CP12 C23A mutant. Increasing evidence suggests that the CP12-PRK interaction is not dependent on the formation of the N-terminal disulfide bridge (Figure 5b,c). Indeed, in the CP12 proteins from the cyanobacterium *S. elongatus PCC7942* and the red alga *Galderia sulphuraria*, the N-terminal cysteine residues are missing [9], but the ternary complex exists [14,41]. In the available crystallographic structures of the *A. thaliana* GAPDH-CP12-PRK complex [29], the CP12 N-terminal cysteine residues pair does not participate in the CP12-PRK interaction (Figure 5b), suggesting that the CP12 N-terminal disulfide is not involved in the PRK protection against irreversible damage. All these arguments rather suggest that CP12-PRK interaction is mostly dependent on the structuration of the A34–H47 region into an α-helix. The crystallographic structure of light-active PRK indicated that the groove of the active site is large enough to accommodate the A34-H47 helix, with a positively charged surface (in blue) facing the negatively charged surface (in red) of the helix (Figure 5c). Besides, we have shown that, in the reduced state, a small proportion of pre-formed helix exists in the disordered CP12, though in very low proportion (below 20%) [10]. We hypothesized that the presence of a small amount of pre-formed helical motifs in reduced CP12 drives the formation of a low affinity complex with PRK [42]. Intrinsically disordered proteins such as CP12 have been shown to form physiologically relevant low affinity complex [43]. PRK shielding against irreversible damage by CP12 could be related to a low affinity complex between these two proteins. This new function for light-reduced CP12 is probably not exclusive [34].

The relative quantitative proteomic analysis showed that the abundance of proteins related to the photochemical phase of photosynthesis was significantly decreased in the absence of CP12, in particular, those of the light harvesting complex. Previous data on *N. tabacum*, *A. thaliana* and *S. guianensis* with deleted CP12 showed that they have a reduced photosynthetic efficiency [21,22,23]. Downstream of the photochemical phase of photosynthesis, the electron flux also seemed reduced because the cytochrome b6 f, the ferredoxin and the NADP^+^ ferredoxin reductase were also in a lower amount in the ΔCP12-Cr line. The CBB cycle is dependent on reduced NADPH produced from the photochemical phase; two key enzymes that catalyze irreversible reactions of the CBB decreased: the PRK and the FBP−2. The redox regulation of CBB enzymes is mediated by thioredoxin F1 among others [44]; it also decreased in ΔCP12-Cr cells.

One of the proteins that was the most increased in ΔCP12-Cr is the intrinsically disordered EPYC1 protein, which clusters RuBisCO in the pyrenoid at low CO_2_ concentrations [45,46]. This is a key actor for the constitution of the pyrenoid as a separate liquid phase or membrane-less organelle—however, in our conditions at 2% CO_2_ concentration, we do not expect the presence of a pyrenoid. Indeed, photorespiration enzymes were not differentially expressed (except the glycerate kinase) [47], confirming that ΔCP12-Cr cells were not CO_2_ limited. Interestingly, RuBisCO LSMT was present at a lower amount in ΔCP12-Cr compared to WT-Cr. RuBisCO LSMT methylates the Lys at position 14 on the large subunit of RuBisCO [48]. A down-regulation of the CBB aldolase has been reported upon methylation [49]. Though the impact of this methylation on RuBisCO activity remains unknown to the best of our knowledge, the lower amount of the RuBisCO LSMT could be related to a modification of RuBisCO regulation. In the context of a lower amount of PRK, a reduction in the RuBisCO substrate, RuBP, is also anticipated. Analogous to a deficiency of CO_2_, deficiency of RuBP may be compensated by an adapted regulation of methylated RuBisCO that could be sequestrated by EPYC1 [50].

In *C. reinhardtii*, products of the CBB can feed: (i) the gluconeogenesis pathway to produce starch, (ii) the glycolysis pathway that promotes fatty acid synthesis [35] or (iii) the oxidative pentose phosphate pathway that is inhibited in the light (reducing conditions) in the chloroplast to avoid futile cycle [3]. In the absence of CP12, two enzymes of the gluconeogenesis pathway were slightly more abundant than in the presence of CP12; the starch synthase was less abundant. One enzyme of the glycolysis pathway was less abundant, suggesting that CP12 is involved in the gluconeogenesis/glycolysis balance. Our results confirm that the algal CP12 is a key regulator of the carbon partitioning, as shown in higher plants [20]. Level of glucose−6-phosphate and its metabolization by carbohydrate pathways are important regulators for nitrogen uptake in *A. thaliana* plants [51], together with the redox homeostasis [52]. In *C. reinhardtii*, a decrease of the nitrogen metabolism is also observed upon deletion of CP12; this could be an indirect consequence of the induced modification of the carbon metabolic partitioning, redox homeostasis and increase of ROS-protective enzymes.

## 4. Materials and Methods

### 4.1. Algal Strains and Culture Conditions

*C. reinhardtii* CC4349 (purchased from Chlamydomonas Resource Center, University of Minnesota, Minneapolis, MN, USA) was maintained in Tris-Acetic acid-Phosphate (TAP) medium supplemented with agar (15 g·L^−1^) and Hygromycin (30 µg·mL^−1^) for the transformant lines. These cells are cell-wall-less and widely used in *C. reinhardtii* fields. It is on this strain that the CRISPR-Cas9 method has been developed [53,54].

To generate knock-out ΔCP12-Cr mutants, CRISPR-Cas9 RNP based genome editing method was used as described previously with a few modifications. To increase the efficiency and easy selection of knockouts, the gene disruption by insertion of exogenous DNA (knock-in) was combined. Therefore, the *aph7* gene expression cassette was co–transformed with the CRISPR/Cas9-RNP complex. The sgRNA sequence for the mutant implementation using CRISPR-Cas9 was designed by Cas-Designer (http://www.rgenome.net/cas-designer/, accessed on 1 October 2019). In the *CP12* gene (Cre08.g380250), the sgRNA target was selected as 5′-CCA TGC AGC CCG CTG CGA GCC GGC AA−3′. The sgRNA was synthesized using GeneArt™ Precision gRNA Synthesis Kit (Invitrogen, Carlsbad, CA, USA). DNA fragment for knock-in, *aph7* gene (aminoglycoside phosphotransferase7, resistance against hygromycin B) was amplified by PCR (1.6 kb) and was purified by agarose gel extraction. The 400 × 10^4^ cells were incubated with the RNP complex and 1 μg of *aph7* for 5 min and transformed with a Gene Pulser Xcell Electroporation System (Bio-Rad, CA, USA). After transformation, the antibiotic resistance colonies were obtained. These pre-selected colonies were further subjected to PCR with specific primers adjacent to sgRNA target sites in *cp12* (F: 5′-GCT GAC CGT CTT CAG CGG CCT A−3′, R: 5′-TTG TGC AAG CGA AGG GAG CGG T−3′) for confirming the targeted mutation of the *cp12* gene. ΔCP12-Cr lines were finally obtained.

### 4.2. Growth Measurements

WT-Cr and ΔCP12-Cr cells were grown in 20 mL TAP medium (supplemented with 30 µg·mL^−1^ hygromycin for ΔCP12-Cr) under 50 µmol photons·m^2^·s^−1^ 8.5 h dark/15.5 h light at 23°C and 110 rpm agitation. A few mL was added in a hermetically closed bottle of 500 mL with 200 mL MOPS culture medium to reach an OD at 680 nm of 0.05. The MOPS culture medium was: 20 mM MOPS, 1 mM KPi, Beijering Salts (7.5 mM NH_4_ Cl, 0.4 mM MgSO_4_ and 0.34 mM CaCl_2_) and Hutner’s trace elements [55] at pH 7.4. Cultures were grown under continuous light, 120 µmol photons·m^2^·s^−1^, 110 rpm at 25 °C; 2% CO_2_ were bubbled during growth via El-Flow Bronkhurst manometres.

Concerning growth monitoring under dark/light conditions, a few mL were added from continuous light culture to reach an OD at 680 nm of 0.1. WT-Cr and ΔCP12-Cr were grown phototrophically under 50 µmol of photons·m^2^·s^−1^, 8.5 h dark / 15.5 h light, at 23 °C, 110 rpm and 2% CO_2_.

The culture density was monitored by absorbance at 680 nm; the number of cells was counted using a Neubauer cell on a microscope Motic BA310 E.

The growth curves were fitted to the following equation [24]:(1)$$Number\;of\;cells=\frac{A}{1+exp^{\frac{4\mu }{A}\times \left(\lambda -t\right)+2}}$$
where *A* is analogous to the asymptote, *µ* is analogous to the maximum growth rate; *λ* is analogous to the lag phase time.

### 4.3. Sample Preparation for Relative Quantitative Proteomic Experiments

A total of 10^9^ cells in the stationary phase of the growth were collected in biological triplicates, for the two conditions—the one where CP12 was not expressed (∆CP12-Cr cells) versus the control where it was (WT-Cr cells) —then pelleted and kept in ice. Following the manufacturer Protifi’s protocol for digestion on S-Trap™ Micro Spin columns (Protifi, NY, USA), we extracted the proteins, reduced them by DTT, alkylated them by iodoacetamide and digested them by trypsin/Lys C protease, including 0.025% ProteaseMax Surfactant Trypsin Enhancer (*v*/*v*), as previously described [56].

Tryptic peptide aliquots of 1 µg were used for LC-MSMS analysis on an ESI-Q-Exactive Plus mass spectrometer (Thermo Scientific, San Diego, CA, USA) coupled to a nano liquid chromatography Ultimate 3000 (Thermo Scientific). Separation by LC was performed on an analytical C18 reversed phase EASY-Spray column (PepMaP^TM^ RSLC, C18, 2 µm, 100 Å, 75 µm ID × 50 cm, Thermo Fisher Scientific), using a three step-linear gradient from 2% to 30% of mobile phase B (0.1% *v*/*v* formic acid (FA)/ 80% *v*/*v* acetonitrile) in mobile phase A (0.1% *v*/*v* FA) for 90 min, a second step from 30% to 50% of B in A for 17 min, and a third step from 50% to 80% in 8 min, followed by a chase step at 95% of B and an equilibration of the column at 2% of B. Mass spectral data were acquired using a top 10 data-dependent acquisition mode with a dynamic exclusion of 30 s, in a 350–1900 *m*/*z* range for the full MS (resolution 70,000 at *m*/*z* 200), and MS/MS acquisition at resolution 17,500 at *m*/*z* 200. For peptide ionization in the nanosource EASY-Spray, voltage was set at 1.9 kV and the capillary temperature at 250 °C. 

### 4.4. Relative Quantitative Proteomic Data Analysis

The data were analyzed by the MaxQuant computational proteomics platform (version 1.6.5.0) used for protein identification and quantification [57] and the Perseus statistical platform (version 1.6.5.0) [58], as previously described in [56]. The database of *C. reinhardtii* v5.6 extracted from Phytozome v13 (19526 entries) was used for the search, supplemented with a set of 245 frequently observed contaminants. The following criteria were used for peptide search: (i) trypsin cleavage authorized before proline with two missed cleavages allowed; (ii) monoisotopic precursor tolerance of 20 ppm in the first search used for recalibration, followed by 4.5 ppm for the main search and 0.5 Da for fragment ions from MS/MS; (iii) cysteine carbamidomethylation (+57.025) as a fixed modification and methionine oxidation (+15.999), serine, threonine and tyrosine phosphorylation (+79.996) and N-terminal acetylation (+42.011) as variable modifications; (iv) a maximum of five modifications per peptide allowed and (v) minimum peptide length was 7 amino acids and a maximum mass of 4600 Da. Spectral alignment was performed; the “Match between runs” option allowed for the transfer of identifications between LC-MS/MS, based on the m/z and the retention time by using the default settings. The false-discovery rate (FDR) on the identification at the peptide and protein levels was set to 1% and determined by searching a reverse database. For protein quantification, unique and razor peptides were used, and considered irrespectively of their modification state mentioned above. The statistical analysis was carried out with the Perseus program. The LFQ normalized intensities were transformed by a base logarithm of 2 to obtain a normal distribution. A minimum of one peptide without any modification was considered to identify and quantify a protein. Quantifiable proteins (valid values) were defined as those detected in 66% of the total group (WT and ∆CP12): 2524 proteins were kept for the statistical test without replacing missing values. To determine whether a given detected protein was present in significantly different amounts, a two-sample *t*-test was applied using a permutation-based FDR-controlled at 0.01 or at 0.05 (250 permutations). The results were illustrated in a Volcano plot, where p values (the probability that the difference in abundance is null) are plotted against the difference in protein amount. The p value was adjusted using a scaling factor s0 to a value of 0.1.

### 4.5. Activity Measurements

Cells (10^9^) of *C. reinhardtii* WT or ∆CP12 strains were centrifuged for 10 min at 2000 g and resuspended in 20 mL of lysis buffer (20 mM Tris pH 7.9, 4 mM EDTA, 5 mM cysteine, 0.1 mM NAD in the presence of protease inhibitors (0.5 µg mL^−1^ cocktail, Sigma Inc., Saint Louis, MO, USA)). Cell lysis was performed by sonication (Sonic Ruptor 250, on ice, 4 cycles, 1 min sonication and 1 min rest) and the membrane fraction was removed by 10 min centrifugation at 10,000 g. Only the soluble fraction (hereafter mentioned as cell extract) was used for activity assay.

GAPDH activity was monitored with 0.2 mM NADPH plus 1 mM of 1, 3-bisphosphoglycerate (BPGA) that was synthesized according to [59]. PRK activity was measured as previously described [60]. Both activities were followed with NAD(P)H disappearance at 340 nm, using a Lambda 25 UV Vis spectrophotometer (Perkin Elmer, PTP−6 Peltier System). All activities were measured in the presence of 5 mM dithiothreitol (DTT) in the assay cuvette. Activity was measured in three biological replicates, both for the WT-Cr and for the ∆CP12-Cr strains. All reagents for enzyme activity measurements were from Sigma Inc. Crude extracts were also incubated on ice for 20 min with 20 mM DTT, before PRK and GAPDH activity were measured. Aliquots from the mixtures were withdrawn after 20 min, and the activities of PRK and GAPDH were measured. For untreated GAPDH, reduced and untreated PRK, one technical replicate was analyzed for each biological replicate, while two technical replicates were analyzed for each biological replicate for reduced GAPDH.

### 4.6. Progress Curve Analysis

Curve fittings were performed with Sigma Plot software (v. 14.0, Systat Software GmbH, Erkrath, Germany) using the following equation based on the one given by Neet and similar to the one used by Lebreton et al. [61,62]:(2)$$P=p\left(1\right)\times t\left[\frac{p\left(1\right)-p\left(2\right)}{k}\left(1-e^{-kt}\right)\right]$$
where *P* is the product appearance, *p*(1) and *p*(2) stand for the stationary-state and the pre-steady-state apparent rate constant, respectively, and *t* is the reaction time. The parameter *k* represents the interconversion of a less active form present during the lag phase into a more active form present in the steady-state [61,62]. When the curves were linear, fitting was performed using the equation of a straight line.

### 4.7. Assay of PRK Protection against Irreversible Inhibition by CP12

Recombinant PRK (0.09 nmol (dimer)) was incubated in the presence of wild-type CP12 (0.18 nmol) in a reconstitution buffer (30 mM Tris-HCl, pH 7.9, 4 mM EDTA, 0.1 mM NAD and 5 mM cysteine) in a 30 mL final volume for three days at 4 °C. PRK activity was measured, at different days, as previously described, but using 5 U of spinach phosphoribose isomerase (Sigma) [60] in the mixture and compared with the PRK alone (control). All activities were measured without addition of dithiothreitol (DTT) in the assay cuvette. Recombinant PRK was obtained as described previously [36].

## 5. Conclusions

Deletion of CP12 in *C. reinhardtii* has direct and/or indirect consequences on key cellular processes such as photosynthesis, redox homeostasis, carbon allocation, mineral nutriments import and proteostasis, as indicated by quantitative proteomics. Above all, our results showed that CP12 is a master of the CBB cycle; beyond this role, it is also able to protect PRK from irreversible inactivation. These findings highlighted CP12 as moonlighting functions of CP12 which are still under investigations. Nevertheless, deletion of CP12 does not impair optimal growth in our present culture conditions, which suggests a re-routing of the microalgae metabolism. The decrease in PRK amount and its lower activity are, thus, possibly compensated by specific cellular adaptations. To conclude, despite its small size, CP12 plays diverse and important roles in algal metabolism.

## Figures and Tables

**Figure 1 ijms-23-02710-f001:**
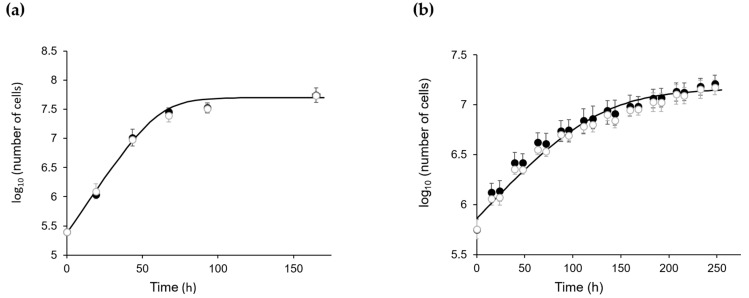
Growth of ΔCP12-Cr (white circle) and WT-Cr (black circle). (**a**) photo-autotrophic and continuous moderate light conditions, 120 µmol of photons·m^2^·s^−1^, 23 °C, 110 rpm and 2% CO_2_. (**b**) photo-autotrophic and dark/light conditions: 50 µmol of photons·m^2^·s^−1^, 8.5 h dark / 15.5 h light, 23 °C, 110 rpm and 2% CO_2_. Growth curves were fitted to data point with Equation (1) (Materials and Methods) [24].

**Figure 2 ijms-23-02710-f002:**
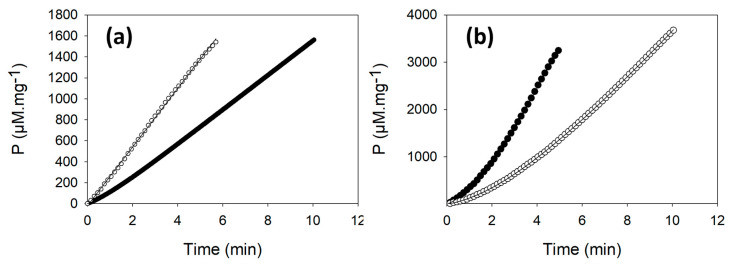
Representative progress curves of appearance of the product of the NADPH dependent GAPDH (**a**) and PRK (**b**) activities. The activity of the two enzymes was measured in the crude extract of WT-Cr (full circles) or of ∆CP12-Cr (empty circles) cells without pre-treatment with DTT. P corresponds to the product concentration in µmol·L^−1^ per mg of total proteins (µM·mg^−1^) in cell extracts. The quantity of proteins from WT-Cr and ΔCP12-Cr cell extracts used in the GAPDH activity assays were, respectively, 84 and 64 µg. In the shown PRK activity assays, 43 and 34 µg of proteins from WT-Cr and ∆CP12-Cr cell extracts were used. The experimental results are fitted to Equation (2) (Materials and Methods), except for the NADPH dependent GAPDH activity progress curve from the ∆CP12-Cr cell extract that is fitted to a straight line.

**Figure 3 ijms-23-02710-f003:**
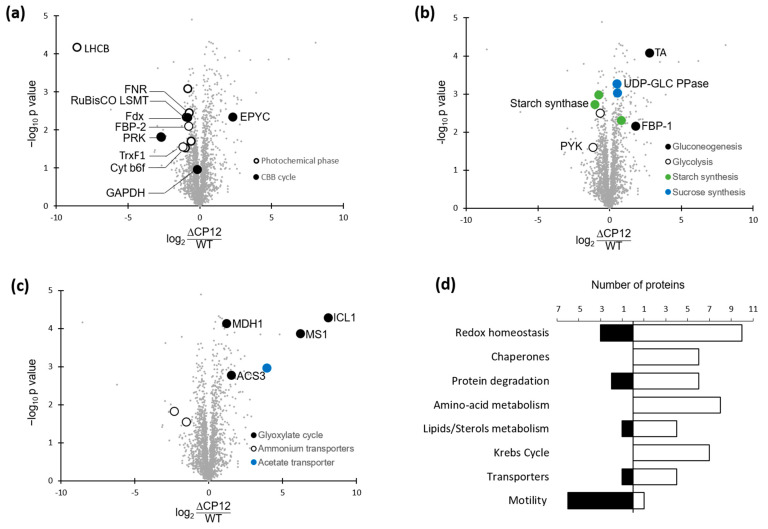
Differentially expressed proteins in ΔCP12-Cr vs WT-Cr strain. (**a**) Plot highlighting proteins related to photosynthesis metabolic pathways. The plot represents the fold changes in protein abundance. The x-axis shows difference of protein abundance in ΔCP12-Cr vs Wt-Cr (in log_2_). The y-axis shows the -log_10_ p value. The Volcano plot is available in Appendix A. (**b**) Plot highlighting proteins involved in carbohydrates related pathways. (**c**) Plot highlighting other metabolic pathways. (**d**) Number of proteins increasing or decreasing in abundance in ΔCP12-Cr strain, classified by selected metabolic pathways. The proteins that are more abundant in ΔCP12-Cr compared to WT-Cr are counted in the white bars, the proteins that are less abundant are counted in the dark bars. All the metabolic pathways are shown in Appendix A. LHCB, light-harvesting chlorophyll a/b-binding; FNR, Ferredoxin NADP^+^ Reductase; RuBisCO LSMT, RuBisCO large subunit methyltransferase; Fdx, ferredoxin; FBP, Fructose-1,6-bisphosphatase; TrxF1, Thioredoxin F1; Cyt b6 f, Cytochrome b6 f complex; EPYC, essential pyrenoid component 1; TA, transaldolase; UDP-GLC PPase, UDP-glucose pyrophosphorylase; PYK, pyruvate kinase; ICL1, Isocitrate Lyase 1; MS1, Malate synthase 1; MDH1, Malate dehydrogenase 1; ACS3, Acetyl-CoA synthetase 3.

**Figure 4 ijms-23-02710-f004:**
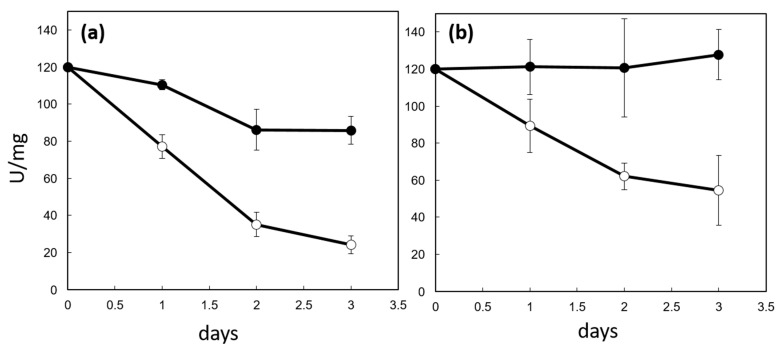
Effects of CP12 on PRK activity. Mixtures (1:1 molar ratio) of CP12 and PRK (black circle) and PRK (white circle) were incubated at 4 °C and PRK activity was measured at four time points (**a**). The same samples were also reduced with 20 mM DTT for 1 h prior activity measurement (**b**). The PRK activity is reported as units per mg of PRK. One unit (U) is the amount of enzyme that catalyzes the reaction of one µmol of substrate per minute.

**Figure 5 ijms-23-02710-f005:**
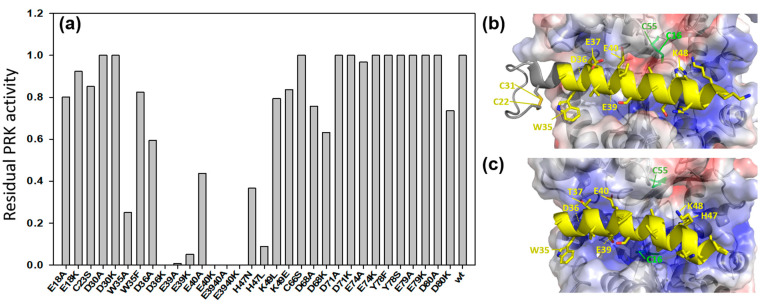
(**a**) Residual activity of PRK recovered after incubation with various CP12 mutants during three days at 4 °C (the absolute PRK activity without incubation is reported in Figure 4 and was 120 U·mg^−1^). The PRK activity was measured and compared with PRK activity of the enzyme incubated without CP12. Residual activity was determined as the ratio of the difference between PRK activity of the PRK-mutated CP12 mixture and of PRK alone. (**b**) Crystallographic structure of oxidized and inactive PRK in complex with CP12 in the ternary complex of *A. thaliana* GAPDH-CP12-PRK (pdb: 6 KEZ) [29]. (**c**) Structural model of A34–H47 helix from the *C. reinhardtii* CP12 (DOI: 10.5452/ma-cr3 y1) [32] docked onto the crystallographic structure of reduced and active *C. reinhardtii* PRK (pdb: 6 H7 G; [30]), based on the *A. thaliana* ternary complex (pdb:6 KEZ) as template. Blue: positively charged surface. Red: negatively charged surface.

**Figure 6 ijms-23-02710-f006:**
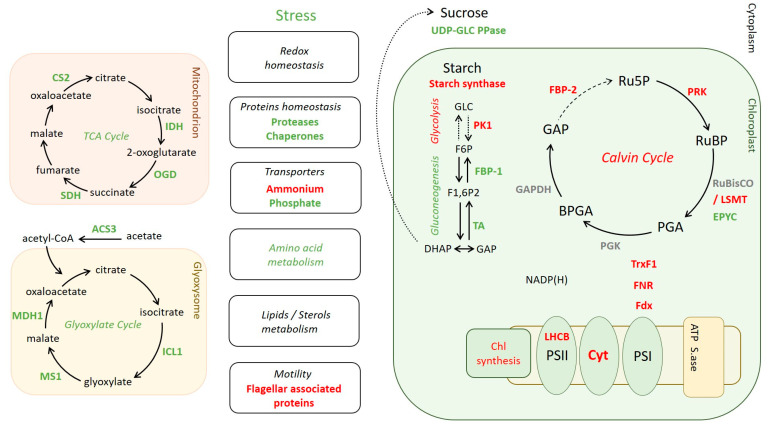
Effect of CP12 deletion on *C. reinhardtii* protein amounts. Green: higher amount in ΔCP12-Cr compared to WT-Cr. Red: lower amount in ΔCP12-Cr compared to WT-Cr. Gray: no change. LHCB, Light-harvesting chlorophyll a/b-binding; Cyt, Cytochrome b6 f; PS, photosystem; ATP S.ase, ATP synthase; Fdx, ferredoxin; FNR, Ferredoxin NADP ^+^ reductase; TrxF1, thioredoxin F1; PGK, Phosphoglycerate kinase; BPGA, 1,3-bisphosphoglycerate; GAP, Glyceraldehyde−3-phosphate; GAPDH, Glyceraldehyde−3-phosphate dehydrogenase; PGA, 3-phosphoglycerate; FBP, Fructose−1,6-bisphosphatase; Ru5 P, Ribulose 5-phosphate; PRK, phosphoribulokinase; RuBP, Ribulose−1,5- bisphosphate; RuBisCO LSMT, RuBisCO large subunit methyltransferase; EPYC, essential pyrenoid component; UDP-GLC PPase, UDP-glucose pyrophosphorylase; PK1, Pyruvate kinase 1; TA, transaldolase; F6 P, fructose−6-phosphate; F1,6 P2, Fructose−1,6-bisphosphate; DHAP, Dihydroxyacetone phosphate; IDH, Isocitrate dehydrogenase; OGD, 2-oxoglutarate dehydrogenase; SDH, Succinate dehydrogenase; CS2, citrate synthase 2; ACS3, Acetyl-CoA synthetase 3; ICL1, Isocitrate Lyase 1; MS1, Malate synthase 1; MDH1, Malate dehydrogenase 1.

**Table 1 ijms-23-02710-t001:** NADPH-dependent GAPDH and PRK activity per mg of total proteins in cell extracts treated with a reducing agent and in untreated cell extracts of ΔCP12-Cr and WT-Cr strains. Standard deviation of three biological replicates is indicated except for GAPDH activity in reduced cell extracts for which standard deviation of three biological replicates plus two technical replicates are indicated. The total protein amount in the PRK activity assays ranged from 27 to 60 µg WT-Cr cell extracts and from 9 to 35 µg ΔCP12-Cr cell extracts. The total protein amount in the GAPDH activity assays ranged from 54 to 120 µg for WT-Cr cell extracts and from 18 to 70 µg for ΔCP12-Cr cell extracts.

	NADPH-Dependent GAPDH Activity (µM.min^−1^)		PRK Activity (µM.min^−1^)	
	Reduced Cell Extract ^1^	Untreated ^2^	Reduced Cell Extract ^1^	Untreated ^2^
WT-Cr	212 ± 73	168 ± 47	1143 ± 328	940 ± 309
ΔCP12-Cr	332 ± 47	320 ± 61	430 ± 94	395 ± 99

^1^ Cell extracts were reduced with 20 mM DTT for 20 min before the assay. ^2^ The activity corresponds to the linear phase of the progress curves.

## Data Availability

The mass spectrometry proteomics data have been deposited to the ProteomeXchange Consortium via the PRIDE [63] partner repository (http://www.ebi.ac.uk/pride) with the dataset identifier PXD031803, accepted on the 19 February 2022.

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
