# Peer review of "Reduction in Phosphoribulokinase Amount and Re-Routing Metabolism in Chlamydomonas reinhardtii CP12 Mutants"

_ijms, 2022, doi:10.3390/ijms23052710_

Round 1

Reviewer 1 Report

Contribution to the field

The small chloroplast protein CP12 is a regulator of the Calvin Cycle, widespread across photosynthetic organisms. It forms a ternary complex with two enzymes (GAPDH and PRK), inhibiting progression of the cycle in the dark, when light-dependent reaction are quiescent. There is a growing body of evidence that it performs additional functions. In the present study, the gene was knocked out in the model alga Chlamydomonas reinhardtii, and the effects were assessed in terms of growth, proteome remodelling, and activity of partner enzyme. The most significant contribution is perhaps in terms of the molecular binding with PRK, which is elegantly demonstrated with numerous site-directed mutants. The proteomic work leads to a several of interesting hypotheses regarding C-partitioning which can be followed-up with metabolic experiments. Some experimental choices need to be better substantiated, and if additional data pertaining to these questions were available, it should be included in SI. Data/discussion around Table 1 appears to be problematic unless we misunderstood it.

Experimental choices

  • L73: rationale for choosing a wall-less strain, which does not allow crossing?
  • Growth (Fig. 1) was seemingly monitored in CO2-enriched air, instead of air. Could this not have masked differences in the growth phenotype (see also L325-327)?
  • Fig. 2: why were all assays not performed with the same amount of total proteins? Is the plotted data normalized?
  • L552-554: why different (n)?
  • Why was CP12 knock-out not validated with western blot, when there is a risk that CRISPR-Cas9 produces an in-frame deletion. qRT-PCR would also have been a suitable validation of the sequencing data.
  • Why was protein accumulation of GAPDH and PRK not cross-validated with a western blot? Anti-PRK for example is available commercially.
  • Fig. 5a: consider adding a threshold to the figure. E40A has a higher activity than H47L but the latter is not listed L273-274.

Request for revision

Reported p-values from t-test and significance of the results. The high variances and low (n) in terms of biological replicates do not always support significant difference. In sections 2.2, a t-test on the 1.3-fold decrease of the WT (L114) returns a non-significant p-value around 0.3, yet 0.01 is reported. In Section 2.3 (L140-141), the treated-untreated fold difference for the WT is 1.2 and not 2.7 as reported, and a t-test gives an insignificant p-value. Data and statistical should be revisited.

Minor corrections

  • Line 18: delete duplicate “in”.
  • Lines 29-30: inconsistent use of separators (comma and semi-colon).
  • Line 41: check spacebar/white after comma.
  • Line 44: replace “Under dark” with “In the dark”.
  • L108: incorrect use of “biphasic curve”.
  • L157: 0.01 (dot not comma separator).
  • Caption to figure 3: inconsistent use of colons after (i), missing comma after TrxF1, remove all spaces before semi-colon (also in caption to Fig. 6).
  • L341: remove comma after “we”.
  • Check italicization of species names throughout the manuscript.

Reviewer 2 Report

This work explores the consequences of knocking out the regulatory chloroplast protein CP12 that functions as regulator of Calvin cycle enzymes in Chlamydomonas. This is the first attempt to explore the protein function in a green model alga. The paper is well written, in part quite wordy. There is novelty in the data. Revision and clarification are needed.

Growth rate was unaffected, however several interesting changes in the cell proteome were detected..

The statistical description of the experiments and data is missing, e.g. in Table 1. How often was it done? Do you indicate SE or SD? Was significance of difference calculated, with which test and which posthoc analysis? All this information must be contained in the legends. Students T-test is insufficient. What is µM do you mean µmol or µmol/L? What was the reference parameter? Cell number, protein contents? Please indicate.

The untargeted proteomics part is interesting. A set of 130 polypeptides was altered in abundance in cp12 vs. wildtype. How can the difference between activity determination and protein amount be explained, namely a 3-fold decrease in PRK activity, but a 6-fold in protein amounts. How specific is the photometric test?

The enumeration of the changes proteins and functions should be shortened. May be you could indicate in the text in brackets how many peptides you found for each polypeptide and comment whether all of them responded accordingly?

The denomination of the x- and y-axis should be completed and the legends should contain all information needed to understand the figures and graphs.

The site-directed mutation part is novel and interesting. However Fig. 5 displays relative activities. Please provide absolute activities as well. Were all mutant variants similarly active?

The discussion is quite lengthy and should be shortened and streamlined. E.g. the long discussion of the glyoxylate pathway without addition evidence may be shortened.

The heading of Fig. 6 appears inadequate since you did not measure metabolites; please change to ‘Effect of CP12 deletion on transcript amounts of enzymes in C. reinhardtii metabolism’

You speculate that CP12 protects PRK; this assumption rests on the prerequisite that sufficient amounts of CP12 are present in the stroma. Please comment on this, also in the manuscript. You have quantitative protein data. Does the stoichiometry of CP12 and PRK support your assumption?

In the proteomics section, hopefully I didn’t miss it, the criteria were not mentioned that had to be fulfilled to claim reliable identification of a peptide. Usually, several peptides should be found and show similar responses to the lack of CP12. Please explain in t M&M. Reference to previous paper is not sufficient.

  1. 18: in in
  2. 24: amino acid residues in the CP protein
  3. 26: instead of ‘decreased by deletion’ … ‘decreased in the absence of CP12’.
  4. 27: functions instead of roles (role twice)
  5. 36: post-translational redox modifications … metabolite concentrations
  6. 74: using the method of CRISPR-Cas9 ribonucleoprotein (RNP)-mediated knock-in (this sentence is difficult to understand because you are talking first about knock-out and then about knock-in: please explain to the reader)
  7. 96: What does ‘continuously’ mean? You mean a time series, continuously is different. This is unclear. Does it mean that you determined the activity in a spectrophotometer as time-dependent recording of absorbance?
  8. 106 ff: How do you know that the lag phase is due to dissociation of the complex; experimental proof should be given; or this interpretation should be moved to the discussion because it is not an own result.
  9. 123: Progress curve is an uncommon description of the graph, isn’t it. The denomination of the y-axis is unclear. Again: What is µM, a concentration? Shouldn’t it be an amount [mol] per mg (what kind of reference: protein? Yes mentioned in text, please give full description in the figure). Why were different amounts of protein added?
  10. 137: You speculate about the complex again. Please separate interpretation from results.

Figure 3: What was the variation in protein amounts between the samples?

  1. 236: Which aldolase is involved in converting linolenic acid to jasmonic acid? Please explain.

Figure 4: Please give denomination of y-axis in full; unit is not a SI unit. Addition of TRX-f1 would be interesting to be tested in the presence and absence of DTT.

  1. 317: As in cyanobacteria,
  2. 575: The term nutriments is uncommon in plant science, do you mean mineral nutrients?

Avoid the word confirm, because it means that there is no novelty?
